# Time Trends in Age at Menarche and Related Non-Communicable Disease Risk during the 20th Century in Mexico

**DOI:** 10.3390/nu11020394

**Published:** 2019-02-13

**Authors:** Inga Petersohn, Arli G. Zarate-Ortiz, Ana C. Cepeda-Lopez, Alida Melse-Boonstra

**Affiliations:** 1Division of Human Nutrition and Health, Wageningen University and Research, 6708WE Wageningen, The Netherlands; inga.petersohn@outlook.de (I.P.); arli.zarateortiz@wur.nl (A.G.Z.-O.); 2Health Sciences Division, Universidad de Monterrey, San Pedro Garza García, N.L. 66238, Mexico; ana.cepeda@udem.edu

**Keywords:** menarche, body mass index, height, diabetes mellitus, non-communicable disease

## Abstract

Developed countries have shown a time trend towards a younger age at menarche (AAM), which is associated with increased risk of later obesity and non-communicable diseases. This study aimed to assess whether a time trend in AAM is associated with disease risk in Mexican women (*n* = 30,826), using data from the Mexican National Health Survey (2000). Linear and log binomial regression was used for nutritional and disease outcomes, while Welch–ANOVA was used to test for a time trend. AAM (in years) decreased over time (*p* < 0.001), with a maximal difference of 0.99 years between the 1920s (13.6 years) and 1980s (12.6 years ). AAM was negatively associated with weight (β = −1.01 kg; 95% CI −1.006, −1.004) and body mass index (BMI) (β = −1.01 kg/m^2^; −1.007, −1.006), and positively with height (β = 0.18 cm; 0.112, 0.231). AAM was associated with diabetes (RR = 0.95; 0.93, 0.98) and hypercholesterolemia (RR = 0.93; 0.90, 0.95), but not with hypertension, breast cancer or arthritis. In Mexico, AAM decreased significantly during the 20th century. AAM was inversely associated with adult weight and BMI, and positively with height. Women with a later AAM had a lower risk of diabetes and hypercholesterolemia.

## 1. Introduction

It is well known that infancy is a critical period for obesity development, but a growing body of evidence also suggests a relevant role of adolescence due to changes in the amount and distribution of body fat [1,2,3,4]. Moreover, age at menarche (AAM) is a substantial and important period in the health of adolescent girls and has a significant impact on their health in the upcoming years.

AAM is dependent on several intrinsic and extrinsic factors. Low body weight may delay the onset of menstruation [5], while overweight and obesity are closely linked to an earlier pubertal onset and a younger AAM [6,7]. Extrinsic factors such as living conditions, nutrition, overall health, physical activity, and socioeconomic status are important and modifiable aspects in the timing of menarche [8]. As a result, mean AAM differs between developing and developed countries. In developed countries, AAM is low (ranging from 12 years in Greece to 13.5 years in Ireland [8]) and has been constantly decreasing in the last 100 years [9]. In developing and emerging countries, much higher AAMs can be seen (ranging from 12.3 years in Thailand to 16.1 years in Senegal [8]). Here, the trend towards a younger AAM has only become visible in recent decades and at a slower rate [10,11,12].

Among the intrinsic factors that influence sexual maturation, the hormonal signals of the adipose tissue (leptin), pancreas (insulin), and gastrointestinal tract (ghrelin) play an important role in the transmission of metabolic information to the central nervous system to stimulate or delay the onset of puberty [5,13,14]. In line with a recent publication by Reinehr et al., this link between metabolic signals and pubertal onset suggests that pubertal timing may have influence on metabolic health in adult life [4].

Studies suggest a link between AAM and adult weight, body mass index (BMI), and height [15]. A meta-analysis of 48 studies found an inverse association between AAM under the age of 12 years and BMI [16]. Similar results were found in both the Nurses’ Health Study I and II (NSH I and NSH II) [17]. In addition, early AAM was found to be associated with a decreased adult height [15,18], suggesting that AAM is not only influencing weight gain but also linear growth. This is important considering that low height is related to the development of respiratory diseases, lung growth, and respiratory obstruction [18,19,20] as well as non-communicable diseases, including cardiovascular diseases and diabetes [21,22,23]. Moreover, early AAM has also been linked with an increased risk of the development of chronic diseases, such as rheumatoid arthritis and cancer, particularly breast cancer [24,25]. A meta-analysis of 9 studies reported a 3% lower risk of cardiovascular (CV) related death with each one-year delay in menarche [26]. Additionally, several studies have shown an inverse correlation between age at menarche and risk of diabetes and pre-diabetes [17,27,28]. However, smaller sample size studies have failed to reproduce these findings [29]. Further, AAM is suggested to be linked to increased blood triglyceride concentrations [30].

In Mexico, similar to many other countries, a secular trend towards a younger AAM has been observed. Studies conducted in the cities of Xalapa and Mexico City [31] and a rural region of southern Mexico [32] showed a significant decline in the mean AAM in recent decades. To the best of our knowledge, a nationwide research assessing the trend in AAM and its implications for women’s health in Mexico has not been performed. Therefore, in this study, we assessed the presence of a nationwide secular trend of age at menarche from 1902 until 1980. Moreover, we examined possible associations between AAM and nutrition and health outcomes later in life among Mexican women aged between 20 and 98 years.

## 2. Materials and Methods

### 2.1. Design and Study Population

We conducted a secondary cross-sectional study based on data obtained during the Mexican National Health Survey (Encuesta Nacional de Salud—ENSA). Data collection took place from September 1999 until March 2000 (details can be consulted elsewhere) [33]. The number of households was allocated proportionally per urban and rural area. Per state, 14 municipalities were selected with a proportional number of households [33]]. The initial sample consisted of 30,628 women aged >20 years having the required information. Menarche at less than seven years of age or above 19 years was considered to have a pathological reason; therefore, we treated them as missing values [8]. All disease outcomes were checked on plausibility, and any cases exceeding the chosen cut-off points were handled as missing. The cut-off points for the variables weight, height, and BMI were based on a study carried out on the same data by Hernández-Cordero et al. [34], blood pressure measurements were regarded as implausible when being classified as high as a hypertensive crisis, and a cut-off of >5 standard deviations was chosen for blood glucose measurements. After eliminating implausible anthropometric, AAM, and disease outcome data, the final study population comprised 19,215 women. An overview of the complete data cleaning process can be found in Figure 1. The survey protocol was approved by the Ethics Commission of the National Institute of Public Health (INSP by its Spanish acronym), and informed consent was obtained from each participant [33].

### 2.2. Data Collection and Variable Construction

Interviews on current health status took place at home and were led by trained professionals with a background in nursing. Information collected included questions on the prevalence of diabetes, hypertension, arthritis, and breast cancer, as well as on family history of diabetes and hypertension (see Appendix A). On average, each house was visited three times to achieve the required quality and response rate. Next to the questionnaire, weight, height, and blood pressure were measured. Both capillary and venous blood samples (≈19 mL) were taken by trained staff. All blood samples collected to assess blood glucose levels were concentrated and centrifuged in the field laboratory before being sent back to the INSP laboratory [33].

The health needs addressed in ENSA 2000 included variables on health care usage, perceived health, and measurements on non-communicable diseases [33]. The nutritional status of the participants is reflected in the variables height (cm) [18], weight (kg), and BMI (kg/m^2^) [16]. Weight was assessed with a solar scale, calibrated with 10 kg tare, and height was measured by a flexometer and a squad. We selected these variables as relevant outcome variables of timing of menarche based on data availability and literature search. Next to that, occurrence of hypertension [35], diabetes [28], arthritis [25], breast cancer [36], and hypercholesterolemia [35] were considered to be related to timing of menarche and were thus selected for analysis.

The age, sex, area of residence, and socioeconomic status (SES) of the sample women were obtained by means of a household questionnaire. Area of residence was defined as rural for localities with <2500 inhabitants and urban for localities with ≥2500 inhabitants. An SES index was constructed based on housing conditions (flooring and roofing materials); ownership of home appliances (refrigerator, stove, washing machine, television, radio, video player, telephone, and computer); and the number of rooms in the house.

Presence of hypertension was assessed by questionnaire and blood pressure measurements and was defined as systolic blood pressure (sBP >140 mmHg) or diastolic blood pressure (dBP >90 mmHg). The instrument that was used to measure blood pressure was a mercury column sphygmomanometer, model TXJ-10. A subject was considered as diabetic if diagnosed by a doctor or if random blood glucose levels exceeded 200 mg/dL. In order to present descriptive data for different groups of AAM, three categories were formed. Based on the data distribution and in line with literature studied on timing of menarche [16,26,28], “early” menarche was defined as reaching first menstruation before the age of 12 years, while ages of 12–14 years were considered as “normal” and first menstruation after the age of 14 years was defined as “late”.

Missing data were not considered as missing completely at random. Thus, five iterations of multiple imputations were performed for the variables AAM, age, weight and height (to calculate BMI), and family history of diseases. The main independent variables and all dependent variables of the two study questions were treated as predictors in the imputation model. Imputed variables were only chosen for when used as predictors, not as outcomes.

### 2.3. Statistical Analysis

Prior to analysis, we cleaned the dataset by removing outliers. To describe our analytic sample, we stratified women by AAM (early, normal, late). For all continuous outcomes, we calculated means and standard deviations. We present binary outcomes as frequencies and percentages. We considered a *p*-value of <0.05 to be statistically significant. We used multiple linear regressions to analyze the relationship of the nutritional indicators weight, height, BMI, and AAM. The variables of weight and BMI violated the assumption of normality of residuals, and we consequently log transformed them for further analysis. As we expected to find a decrease of AAM in younger individuals, we adjusted the crude model for current age. We used logistic regression models to analyze the association of nutritional status and AAM. The crude model was stepwise adjusted for the confounding variables age, BMI and, if possible, for family history of the disease. As most disease outcomes were expected to violate the rare disease assumption, they were analyzed using log binomial regression with complementary log log link, in order to obtain relative risks (RR). The models used for log binomial regression included only those variables that showed to add to the crude model significantly. Normality of AAM per decade of birth group was visually assessed and all distributions were considered as normal. To study whether AAM differed between the decade groups, we used Welch–ANOVA, because the data failed the assumption of equal variances. The Games–Howell post-Hoc test [37] was performed to further investigate which decade groups differed in statistically significant manner. We performed all of the analyses using IBM SPSS Statistics 23 software (2015. Armonk, NY, USA: IBM Corp).

## 3. Results

### 3.1. Sample Characteristics

Mean AAM of the total study population was 13 years. A total of 4073 women (14 %) were classified as having an “early” AAM, 70% had a “normal” AAM, and 16% were considered “late”. The “early AAM” group was characterized by having the lowest mean current age (37 years), while having the highest BMI (28.8 kg/m^2^) and weight (67.6 kg). The “late AAM” group had the highest mean age (45 years) and the lowest weight (64.2 kg) and BMI (27.3 kg/m^2^). The “late AAM” group had the highest percentage of diseased women for each disease outcome, except for hypercholesterolemia and breast cancer. Hypercholesterolemia prevalence was highest (10%) amongst the “early AAM” group, while breast cancer prevalence was highest in the “early AAM” and “normal AAM” group (4%; Table 1).

The number of subjects per decade group differed from only 53 (1900s) to 8823 in the 1970s. The corresponding standard deviations of the decade groups of 1910s–1980s, however, were comparable. Women born in the 1900s and 1920s had the highest age when reaching menarche, while women born in the 1980s showed the youngest AAM (Figure 2).

### 3.2. Association of Age at Menarche and Nutritional Status

AAM was negatively associated with BMI scores and weight, both before and after adjustment for current age (see Table 2). In the adjusted model, AAM significantly predicted BMI (β= −1.01; 95% CI −1.007, −1.006). The results indicate that the used model explained 4.0% of the variance. Weight was significantly negatively associated with an older AAM (β = −1.01; 95% CI −1.006, −1.004). The combined predictors AAM and current age explained 0.9% of the model variance. In contrast to that, final attained height was positively associated with AAM and current age (β = 0.18; 95% CI 0.112, 0.231). The model explained 6.3% of the variance.

### 3.3. Association of Age at Menarche and Disease Status

Univariable predictor testing proved all possible confounding variables (BMI, age, BMI–age interaction, and family history) to be significantly associated with each disease variable, with the exception of breast cancer. Consequently, the crude models were adjusted for all co-variables mentioned above.

After adjustment, diabetes and hypercholesterolemia were significantly inversely related to AAM with relative risks (RRs) of 0.95 (95% CI 0.93, 0.98) and 0.93 (95% CI 0.90, 0.95), respectively, see Table 3. The adjusted analysis of AAM and hypertension, breast cancer, gout arthritis, and other arthritis showed the hypothesized inverse trend. However, these associations were not statistically significant (*p* > 0.05). AAM was associated with higher disease risk for gout arthritis (RR 1.05); however, this result was not statistically significant (*p* > 0.05) either.

### 3.4. Time Trend of Age at Menarche

Analysis of variances for the association of decade of birth and AAM showed a statistically significant difference between groups (Welch F = 99.42, *p* < 0.001). The largest statistically significant reduction in AAM was seen between the decades of the 1920s and 1980s, in which the mean age of menarche decreased by 0.99 years (see Table 4). A significant downwards trend in AAM between the decades of the 1930s and 1980s was observed.

## 4. Discussion

We observed a trend towards a younger age at menarche and related non-communicable disease risk between the decades of the 1920s and 1980s. We found that earlier AAM was associated with lower adult height, higher adult weight and BMI, and risk of diabetes and hypercholesterolemia.

Our study showed a secular trend towards a younger AAM in Mexico from 1900 to 1980. Mean AAM significantly decreased compared to each previous decade group from the 1930s onwards. Largest total decrease in AAM was between the 1920s and 1980s, with a decline of mean AAM by almost a full year. Marván et al. assessed the declining AAM in the cities of Xalapa and Mexico City. In their study, AAM decreased by 1.6 years between the 1940s (and earlier) and the 1990s [31]. Similar to that, in a rural community in Southern Mexico, a larger decline of 1.8 years took place over the previous 23 years [32]. Thus, our findings indicate a nationwide trend towards a younger AAM, yet there is no consensus on the extent of the decrease. Possible explanations can be the difference in the designs of the studies. Both Marván et al. and Malina et al. used predominately the status quo method to assess AAM, while data in our study were obtained retrospectively. This increased the chance of recall bias and may have decreased the accuracy of our study. Yet, compared to these studies, the study population used in our study was considerably larger and the validity of our results was thus high. Another possible explanation for the difference at hand is the setting in which the studies were conducted. While our study represented the nationwide average of AAM, Marván et al. and Malina et al. focused their research on specific areas, which are geographically and culturally different. Overweight prevalence and poverty rate, as a marker of socioeconomic status, differ between the study locations [38,39]. It is thus suggested that the magnitude of the decrease in AAM differs per location and its specific environmental conditions.

We showed that each one-year delay in menarche was associated with a 0.18 cm increase in final attained height. Our results are consistent with other studies. In the UK, the impact of AAM was greater, with an increase of 0.59 cm in final height per one year of delayed menarche [18]. The EPIC study (European Prospective Investigation into Cancer and Nutrition) assessed the impact of AAM on final height in several European countries in women born before and after 1945. In women born before 1945, there is evidence of an increase in height in France, the Netherlands, and Spain, with 0.41 cm, 0.19 cm, and 0.21 cm, respectively. Interestingly, AAM had a higher impact on women born after 1945 [40]. Pubertal sex hormones and growth hormones generally increase simultaneously, leading to skeletal growth. Height velocity decreases after menarche. On average, final height is attained one year after menarche [41]. Thus, a younger AAM can be linked to an overall shorter stature, which may lead to impaired health outcomes [42,43].

We found adulthood BMI to be significantly reduced with increasing AAM; each one-year delay in menarche was associated with a reduction of 1.01 kg/m^2^. The large US Nurses’ Health Studies (NSH I/NSH II) assessed the years 1980 and 2005 and found a smaller impact of AAM. In the NSH I, a decrease of 0.2 kg/m^2^ was found with each one-year increase in AAM, while in NSH II, a slightly larger reduction (0.26 kg/m^2^) was seen [17]. In Scotland, an earlier study found a decrease in BMI of 0.64 kg/m^2^ [44]. All these findings are in line with our results. However, it is remarkable that the extent to which BMI decreases with increasing AAM in our study is almost twice as high as the findings of Pierce and Leon. Compared to the NSH I, reduction in BMI in our study is five times as high as the decrease found in the US. A higher BMI increases the risk of cardiovascular diseases, diabetes, osteoarthritis, and some cancers [45] and thus represents a large public health burden. Since childhood BMI is closely related to AAM [6,7], the association might be based on reversed causality. Studies suggest that the inverse association of AAM and BMI is not explained by confounding of childhood BMI [44] while in other studies, minor attenuation of results was found [16]. For future studies, we therefore recommend taking childhood BMI into account in the association of AAM and BMI.

Of the non-communicable diseases we assessed, diabetes and hypercholesterolemia were significantly associated with AAM. The risk for diabetes was slightly lower when AAM increased (RR 0.95). This is in line with findings of previous studies that indicated a decreased risk of diabetes with later AAM [17,28,46]. Stöckl and colleagues found an inverse association with an RR of 0.88, which is comparable to the findings of Lakshman et al. who found an OR of 0.91. Data from the NSH II study suggested an RR of 0.97 per one-year delay of menarche [17]. Thus, our findings support the hypothesis of an inverse association of AAM and diabetes risk, found in other studies. Studies on the association of AAM and hypercholesterolemia are rare. However, an association of younger AAM with higher triglyceride concentrations has been observed [30]. No such effect was found in the studies of Stöckl D et al. [28] and Frontini M. et al. [47]. Thus, the specific influence of a young AAM on hypercholesterolemia has not been investigated intensely, while studies on the effect on triglyceride concentrations are inconsistent. Our findings add to the hypothesis of an inverse association between AAM and hypercholesterolemia; however, these results should be validated by more cohort studies investigating hypercholesterolemia directly.

We did not observe a significant association between AAM and risk of hypertension, arthritis or breast cancer. Studies among British [27] and Korean [48] cohorts showed that risk of hypertension decreased with increasing age at menarche, although a more recent study could not reproduce these findings [49]. The lack of association between AAM and risk of hypertension in our population could be explained by the presence of other confounders that were not evaluated in the survey, like physical activity, smoking, alcohol consumption, and educational level. Similarly, the evidence of the effect of early AAM on arthritis has been inconsistent. The NSH I study found a higher risk of rheumatoid arthritis among women with an early AAM (RR: 1.3) [25]. Yet, other studies suggest no association [50,51] or a higher risk with later AAM [52,53]. We observed a decreased risk for breast cancer with later AAM. Although our results were not significant, the observed trend is consistent with another study that showed an increased risk for breast cancer with decreasing AAM [24]. Based on our findings and the current state of the literature, it remains uncertain if AAM is associated with risk of hypertension, breast cancer or arthritis in adulthood.

The strengths of the current study include the large sample size and selection of participants. All participants were chosen randomly and women of all states, covering urban and rural living environments, were included in this study. As a result, the representativeness of our study is expected to be high. Several limitations should also be acknowledged. First, the lack of sociodemographic and lifestyle data that could influence the results. BMI and related non-communicable disease risk are closely associated with a person’s socioeconomic status [54,55] and lifestyle factors such as physical activity and nutritional behavior [56,57]. Socioeconomic status [58], nutrition [59], and physical activity [60] are associated with AAM as well and they thus might act as confounding factors in the studied association. Thus, including these factors in the model might have changed the final results of this study. Lastly, when studying the secular trend of AAM over time, one should consider the differences in sample size. While the decade group of the 1970s consisted of 8823 women, data of only 53 women were available in the decade of the 1900s. Even though standard deviations between all groups are comparable, this difference might explain the non-significant differences of the decades of the 1900s and 1910s. Yet, the overall trend towards a younger age at menarche is not affected by that. Finally, results cannot be interpreted as causality, because of the transversal nature of our study.

In parallel with the obesity epidemic in Mexico, the prevalence of type 2 diabetes mellitus has increased rapidly during the last several decades. It has been shown that by the time type 2 diabetes is diagnosed, some individuals have already developed serious complications. Therefore, it has become increasingly important to identify persons at risk in early life so they may benefit from early interventions. Our findings stress the importance of assessing age at menarche from clinical and public health perspective. Age at menarche can be an indicator of nutritional status and normal development of adolescent girls, and earlier AAM can be an independent risk factor of non-communicable diseases. Because recall bias can be present in retrospective studies, longitudinal and cross-sectional studies in adolescents may help to understand the association between pubertal onset and health. It is recommended to monitor AAM on a regular basis by making it a permanent variable of national health surveys.

## 5. Conclusions

Our research showed a trend towards a younger AAM and related non-communicable disease risk during the 20th century in Mexico. Additionally, we found that younger AAM is associated with lower adult height, higher adult weight and BMI, and higher risk for type 2 diabetes and hypercholesterolemia in adulthood. Thus, early age at menarche might represent a risk factor for early identification of women who are at increased risk of being overweight or obese and of developing type 2 diabetes in adulthood. Yet, in order to confirm these findings, additional research which includes data on sociodemographic and lifestyle factors is needed.

## Figures and Tables

**Figure 1 nutrients-11-00394-f001:**
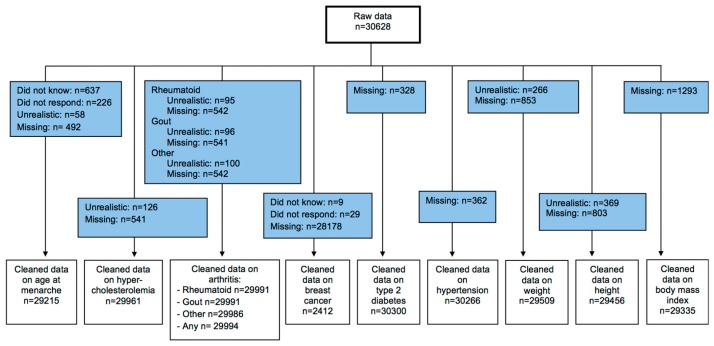
Overview of data cleaning.

**Figure 2 nutrients-11-00394-f002:**
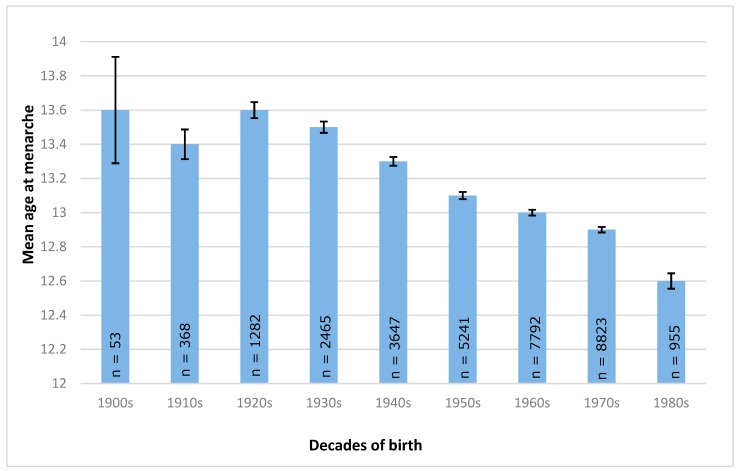
Mean age at menarche per decade of birth (1900s–1980s).

**Table 1 nutrients-11-00394-t001:** Descriptive characteristics of study population per age at menarche group.

	Age at Menarche		
	Early (<12 years)	Normal (12–14 years)	Late (>14 years)
Observed *n*	4238	21333	5055
Observed %	14	70	16
Age (years) ^§^	37 (14.0) ^a^	40 (15.1) ^b^	45 (16.3) ^c^
Height (cm) ^§^	153.3 (7.0) ^a^	152.9 (7.0) ^b^	153.1 (7.2) ^ab^
Weight (kg) ^§^	67.6 (14.1) ^a^	64.9 (13.4) ^b^	64.2 (13.8) ^c^
BMI (kg/m^2^) ^§^	28.8 (5.7) ^a^	27.7 (5.3) ^b^	27.3 (5.3) ^c^
Hypertension ^†^	732 (18%) ^a^	3587 (18%) ^a^	1030 (22%) ^b^
Diabetes ^†^	337 (8%) ^ab^	1559 (8%) ^a^	432 (9%) ^ab^
Arthritis ^†^	201 (5%) ^a^	1122 (6%) ^a^	319 (7%) ^b^
Hypercholesterolemia ^†^	385 (10%) ^a^	1454 (7%) ^b^	374 (8%) ^b^
Breast cancer ^†^	15 (4%) ^a^	70 (4%) ^a^	12 (3%) ^a^

^§^ Results are shown as mean (standard deviations—SDs). ^†^ Results are shown as frequency (% of total) ^abc^ Significant difference between groups, insignificant differences are indicated by same letter (*p* < 0.05).

**Table 2 nutrients-11-00394-t002:** Association between age at menarche and nutritional status.

Outcome Variables	β Estimates	CI	R^2^
Crude model: age at menarche			
BMI	−1.00 *	−1.005, −1.004	0.007
Height	−0.02 *	−0.077, −0.034	0.000
Weight	−1.01 *	−1.005, −1.004	0.006
Adjusted model: age at menarche + current age			
BMI	−1.01 *	−1.007, −1.006	0.040
Height	0.18 *	−0.112, −0.231	0.063
Weight	−1.01 *	−1.006, -1.004	0.009

* *p* < 0.001.

**Table 3 nutrients-11-00394-t003:** Relative risks of the adjusted model.

Disease Outcome	RR ^1^	95% CI ^1^
Diabetes	0.95 *	0.93, 0.98
Hypertension	1.00	0.98, 1.02
Hypercholesterolemia	0.93 *	0.90, 0.95
Breast cancer ^§^	0.95	0.98, 1.02
Arthritis	1.00	0.97, 1.03
Gout arthritis	1.05	0.94, 1.18
Rheumatoid arthritis	1.00	0.99, 1.02
Other arthritis	0.93	0.93, 1.07

^1^ Adjusted for BMI, age, BMI–age interaction and family history. * *p* ≤ 0.001. ^§^ Unadjusted.

**Table 4 nutrients-11-00394-t004:** Mean differences of age at menarche among different decades of birth.

Decade	1900s	1910s	1920s	1930s	1940s	1950s	1960s	1970s	1980s
1900s	1	0.10	0.14	−0.04	−0.20	−0.40	−0.49	−0.69 *	−0.86 *
1910s		1	0.24	−0.13	−0.11	−0.30 *	−0.39 **	−0.59 **	−0.76 **
1920s			1	−0.10	−0.34 **	−0.53 **	−0.63 **	−0.83 **	−0.99 **
1930s				1	−0.24 **	−0.43 **	−0.53 **	−0.72 **	−0.89 **
1940s					1	−0.19 **	−0.29 **	−0.49 **	−0.65 **
1950s						1	−0.10 *	−0.29 **	−0.46 **
1960s							1	−0.20 **	−0.37 *
1970s								1	−0.17 *
1980s									1

* Mean difference significant at the *p* < 0.005 level. ** Mean difference significant at the *p* < 0.001 level.

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
