# Peer review of "Time Trends in Age at Menarche and Related Non-Communicable Disease Risk during the 20th Century in Mexico"

_nutrients, 2019, doi:10.3390/nu11020394_

Round 1
Reviewer 1 Report
This study showed that age at menarche (AAM) decreased significantly during the 20th century in Mexico. AAM was inversely associated with adult weight and BMI, and positively with height. Moreover, women with a later AAM had a lower risk of diabetes and hypercholesterolemia.
This study is well designed and gave us important information that AAM was inversely associated with adult weight and BMI, and positively with height. The article is well-written. I like this article. I made some comments.
1. Food and lifestyle may affect AAM. Could you show the food or lifestyle change during the 20th century? Moreover, could you show how much these food and lifestyle change affect AAM?
2. Could you show the prevalence of diabetes, hypertension, and hypercholesterolemia during 20th century? If the prevalence of them increases, it may affect the results that women with a earlier AAM had a higher risk of diabetes and hypercholesterolemia.
This study has a good impact. I am looking forward to reviewing your revised article.
Author Response
Reviewer 1
Food and lifestyle may affect AAM. Could you show the food or lifestyle change during the 20th century? Moreover, could you show how much these food and lifestyle change affect AAM?
We appreciate the observation from the reviewer. Indeed, lifestyle and food affect age at menarche. The variable AAM in this study was a retrospective question, and unfortunately we do not have data on lifestyle and diet at a pre-pubertal stage from our study subjects. In addition, the National Survey of Health (ENSA-2000) did not assess dietary intake. For that reason, we could not investigate the influence of diet on the association between AAM and health outcomes. However, a sentence in the limitation and conclusion section of the manuscript addresses the importance of including sociodemographic and lifestyle factor that might influence results (line 284-290 & 315-316).
Could you show the prevalence of diabetes, hypertension, and hypercholesterolemia during 20th century? If the prevalence of them increases, it may affect the results that women with an earlier AAM had a higher risk of diabetes and hypercholesterolemia.
We are afraid we do not completely understand the reviewer's comment. The prevalence of obesity, diabetes and hypertension increased among the 20th century while AAM decreased (Barquera et al., 2013; Rivera et al., 2002; Rivera, Irizarry, & Gonzalez-de Cossio, 2009). Our research shows a trend towards a younger AAM and related non-communicable disease risk during the 20th century in Mexico. Additionally, we found that younger AAM is associated with lower adult height, higher adult weight and BMI, and higher risk for type 2 Diabetes and hypercholesterolemia in adulthood.

Reviewer 2 Report
Abstract:
-Lines 13-14- Please rephrase to 'This study aimed to assess whether a time trend in AAM is associated with disease risk in Mexican women (n=30,826), using data from the Mexican National Health Survey (2000).
-Line 15- replace whereas with while
-Spell BMI out in full then use the abbreviation.
Introduction:
-Line 36- Please rephrase to 'As a result, mean AAM differs between developing and developed countries.'
-Lines 37 and 39- (ranging from 12.0 in Greece to 13.5 in Ireland [7])- please indicate the units, I am assuming it should be years?
-Please change 'has been shown to continuously decrease over the last 100 years' to and has been constantly decreasing in the last 100 years'
-Spell BMI out in full then use the abbreviation.
-Lines 41-42- Can you please indicate how many studies were included in the meta-analysis?
-Lines 42-43- Please spell out the abbreviation used.
-Lines 45-47- Please provide further supporting references.
-Lines 47-48- 'Moreover, early AAM has also been linked with an increased risk of the development of chronic diseases.' please provide supporting references for this statement.
-Line 48- Please indicate the number of studies included in this meta-analysis
-I would recommend rephrasing this sentence 'In their meta-analysis on AAM and cardiovascular (CV) death Charalampopoulos, McLoughlin, et al. (2014) found a 3 % lower risk of CV related death with each one-year delay in menarche.' to A meta-analysis of X number of studies reported that a 3 % lower risk of cardiovascular (CV) related death with each one-year delay in menarche.'
-Lines 51-54- This sentence is difficult to read with all the cited author names. Can you please rephrase this sentence so it is easier to read?
-Line 54- Start the sentence with 'However, studies of a smaller sample size have failed... '
-Lines 56-57- Provide supporting references .
-Line 58-60- Please restructure this sentence 'Yet, a nation-wide investigation of a decreasing AAM as well as an indication of its association with adult health in Mexico is lacking.'
-Can you describe the mechanisms know to explain the association between AAM and non-commuincable disease risk?
-Lines 61-63 'Moreover, we examined possible associations between AAM and nutrition and health outcomes later in life among Mexican women aged between 20 and 98 years.' The evidence regarding the association between nutrition and AAM has not really been covered in the introduction.
Methods:
-Lines 73-75- Your in-text referencing used is inconsistent with the previous style used.
-Can you explain how the data was cleaned?
-Lines 75-76- Can you provide further information as to how the plausibility of disease outcomes were checked?
-Were the questions used for the interviews developed specifically for the study or was a validated tool used? Was a standardised interview script used by the interviewers? Can you include the questions as a supplementary file?
- I am a bit concerned it took three house visits to get the health information required. Can you provide further explain as to why this was the case? What extra quality were you trying to achieve?
-What equipment was used and who measured weight, height and blood pressure?
-What was the blood volume of the samples and who collected the samples?
-Line 93- What is 'ENSA 2000'? Please provide further information and spell out the abbreviation.
-Lines 94-95- Please indicate the units of these measures.
-Line 107- Spell out abbreviations
-Do you have a supporting referencing for the Games-Howell post-Hoc test?
-In brackets can you indicate the company and location of the company that produced the statistical software used for the study?
Results:
-Table 1- Please indicate all the units reported (e.g. age, years etc.) and spell out SDs in full in the foot notes
-Table 2- Please indicate all the units reported (e.g. age, years etc.)
-Line 175- please spell out abbreviation RR
Discussion:
- Your in-text referencing used is inconsistent with the previous style used in this section.
Author Response
We are pleased to be given the opportunity to submit a revised version of the manuscript
entitled “Time trends in age at menarche and related non-communicable disease risk during the 20th century in Mexico”. We feel that we have been able to respond to all of the specific points, and we believe this has helped us in improving the manuscript. Below, all comments are listed point-by-point as well as our itemized responses including changes made to the manuscript. Your comments are written in bold, and our reply is written in italics. Changes made to the manuscript are highlighted with the function “track changes” in Microsoft Word.
Reviewer 2
Lines 13-14- Please rephrase to 'This study aimed to assess whether a time trend in AAM is associated with disease risk in Mexican women (n=30,826), using data from the Mexican National Health Survey (2000).
Corrected (lines 12-14)
Line 15- replace whereas with while
Replaced (line 15)
Spell BMI out in full then use the abbreviation.
Has been included (line 22)
Line 36- Please rephrase to 'As a result, mean AAM differs between developing and developed countries.'
Rephrased (line 36)
Lines 37 and 39- (ranging from 12.0 in Greece to 13.5 in Ireland [7])- please indicate the units, I am assuming it should be years?
Corrected (line 37, 39 & 48)
Please change 'has been shown to continuously decrease over the last 100 years' to and has been constantly decreasing in the last 100 years'
Changed (line 38)
Spell BMI out in full then use the abbreviation.
Abbreviation for BMI (body mass index), has been spelled out in the manuscript (line 47)
Lines 41-42- Can you please indicate how many studies were included in the meta-analysis?
Has been included (line 48)
Lines 42-43- Please spell out the abbreviation used.
Abbreviation for NSH I and NSH II, has been spelled out in the manuscript (Nurses’ Health Study I and II) (line 49)
Lines 45-47- Please provide further supporting references.
We now provide additional supporting references for the relationship between low height and the development of respiratory diseases, lung growth and respiratory obstruction as well as non-communicable diseases, including cardiovascular diseases and diabetes (Lines 53-54):
19. Svanes C, Bjørge L, Omenaas ER, Real FG. Respiratory health in women: from menarche to menopause AU - Macsali, Ferenc. Expert Review of Respiratory Medicine. 2012;6(2):187-202.
20. Varraso R, Siroux V, Maccario J, Pin I, Kauffmann F. Asthma Severity Is Associated with Body Mass Index and Early Menarche in Women. American Journal of Respiratory and Critical Care Medicine. 2005;171(4):334-9.
22. Bourgeois B, Watts K, Thomas DM, Carmichael O, Hu FB, Heo M, et al. Associations between height and blood pressure in the United States population. Medicine. 2017;96(50):e9233-e.
23. Janghorbani M, Momeni F, Dehghani M. Hip circumference, height and risk of type 2 diabetes: systematic review and meta-analysis. Obesity reviews : an official journal of the International Association for the Study of Obesity. 2012;13(12):1172-81.
Lines 47-48- 'Moreover, early AAM has also been linked with an increased risk of the development of chronic diseases.' please provide supporting references for this statement.
Supporting references have been included (Line 55):
24. Cancer CGoHFiB. Menarche, menopause, and breast cancer risk: individual participant meta-analysis, including 118 964 women with breast cancer from 117 epidemiological studies. The lancet oncology. 2012;13(11):1141-51.
25. Karlson EW, Mandl LA, Hankinson SE, Grodstein F. Do breast-feeding and other reproductive factors influence future risk of rheumatoid arthritis? Results from the Nurses' Health Study. Arthritis and rheumatism. 2004;50(11):3458-67.
Line 48- Please indicate the number of studies included in this meta-analysis
The number of studies has been included (line 56)
I would recommend rephrasing this sentence 'In their meta-analysis on AAM and cardiovascular (CV) death Charalampopoulos, McLoughlin, et al. (2014) found a 3 % lower risk of CV related death with each one-year delay in menarche.' to A meta-analysis of X number of studies reported that a 3 % lower risk of cardiovascular (CV) related death with each one-year delay in menarche.'
Corrected (lines 56-57)
Lines 51-54- This sentence is difficult to read with all the cited author names. Can you please rephrase this sentence, so it is easier to read?
Thanks for this observation. We have rephrased to: “Additionally, several studies have shown an inverse correlation between age at menarche and risk of diabetes and pre-diabetes [17, 33, 47].” (lines 57-58)
Line 54- Start the sentence with 'However, studies of a smaller sample size have failed... '
Corrected (line 58)
Lines 56-57- Provide supporting references.
To the best of our knowledge, the studies performed by Marvan ML et al. (2016) and Malina RM et al. (2004) are the only available published data looking at trends in age at menarche in Mexico (lines 62-63).
Line 58-60- Please restructure this sentence 'Yet, a nation-wide investigation of a decreasing AAM as well as an indication of its association with adult health in Mexico is lacking.'
Corrected (lines 63-65)
Can you describe the mechanisms know to explain the association between AAM and non-commuincable disease risk?
Thanks for the recommendation. We incorporated to the introduction section (Line 41 - 46): “Among the intrinsic factors that influence sexual maturation, the hormonal signals of the adipose tissue (leptin), pancreas (insulin) and gastrointestinal tract (ghrelin) play an important role in the transmission of metabolic information to the central nervous system to stimulate or delay the onset of puberty. This link between metabolic signals and pubertal onset suggests that pubertal timing may have influence on metabolic health in adult life”.
Lines 61-63 'Moreover, we examined possible associations between AAM and nutrition and health outcomes later in life among Mexican women aged between 20 and 98 years.' The evidence regarding the association between nutrition and AAM has not really been covered in the introduction.
Agree. We now provide evidence regarding the association between nutrition and AAM (Line 41 - 46).
Lines 73-75- Your in-text referencing used is inconsistent with the previous style used.
Corrected.
Can you explain how the data was cleaned?
We complemented the description of data cleaning process (Lines 75-84).
Lines 75-76- Can you provide further information as to how the plausibility of disease outcomes were checked?
Included (lines 79-82)
Were the questions used for the interviews developed specifically for the study or was a validated tool used? Was a standardised interview script used by the interviewers? Can you include the questions as a supplementary file?
1. The questionnaires were developed especially for The National Health Survey (ENSA-2000), and validated in a pilot study. The interviewers were trained and standardized. Details on the methodology can be found in Valdespino OG, López-Barajas MP, Mendoza L, Palma O, Velázquez O, Tapia R, Sepúlveda J. : Encuesta Nacional de Salud 2000. 2000.
2. The original version of the questionnaire has been attached (Spanish version), and the questions of interest have been highlighted.
I am a bit concerned it took three house visits to get the health information required. Can you provide further explain as to why this was the case? What extra quality were you trying to achieve?
Since we only performed a secondary analysis of the Mexican National Health Survey in the first sentence of the methodology section we refer a paper that describes in detail logistics and quality control of the survey (Lines 71-73).
What equipment was used and who measured weight, height and blood pressure?
Thanks for this observation. We agree that this information is important and we now describe it in the methodology section of the manuscript (lines 102-103, 114-116).
What was the blood volume of the samples and who collected the samples?
Details on blood samples collection have been added (Lines 96-98): “Both capillary and venous blood samples (≈19 mL) were taken by trained staff. All blood samples collected to assess blood glucose levels, were concentrated and centrifuged in the field laboratory before being sent back to the INSP laboratory [30].”
Line 93- What is 'ENSA 2000'? Please provide further information and spell out the abbreviation.
Thanks for this observation. The abbreviations for ENSA (National Health survey), has been spelled out (Line 72)
Lines 94-95- Please indicate the units of these measures.
Units for height (m), weight (kg), and BMI (kg/m2) are now included (line 101)
Line 107- Spell out abbreviations
We included abbreviations for systolic (sBP) and diastolic (dBP) blood pressure (line 114).
Do you have a supporting referencing for the Games-Howell post-Hoc test?
Has been included (line 144)
In brackets can you indicate the company and location of the company that produced the statistical software used for the study?
Has been included (Lines 146)
Table 1- Please indicate all the units reported (e.g. age, years etc.) and spell out SDs in full in the foot notes
Units have been included and the abbreviation for SDs (Standard Deviations) has been spelled out.
Line 175- please spell out abbreviation RR
Abbreviation for RR (relative risk) has been spelled out (Line 185).
Your in-text referencing used is inconsistent with the previous style used in this section.
Corrected
Barquera, S., Campos-Nonato, I., Aguilar-Salinas, C., Lopez-Ridaura, R., Arredondo, A., & Rivera-Dommarco, J. (2013). Diabetes in Mexico: cost and management of diabetes and its complications and challenges for health policy. Global Health, 9, 3. doi:10.1186/1744-8603-9-3
Olaiz G, R. R., Barquera S, Shamah T, Aguilar C, Cravioto P, López P, Hernández M, Tapia R, Sepúlveda J. (2003). Encuesta Nacional de Salud 2000. Cuernavaca, Morelos, México Retrieved from https://www.insp.mx/encuestoteca/Encuestas/ENSA2000/OTROS/ensa_tomo2.pdf.
Rivera, J. A., Barquera, S., Campirano, F., Campos, I., Safdie, M., & Tovar, V. (2002). Epidemiological and nutritional transition in Mexico: rapid increase of non-communicable chronic diseases and obesity. Public Health Nutr, 5(1a), 113-122. doi:10.1079/phn2001282
Rivera, J. A., Irizarry, L. M., & Gonzalez-de Cossio, T. (2009). Overview of the nutritional status of the Mexican population in the last two decades. Salud Publica Mex, 51 Suppl 4, S645-656.
Valdespino JL, O. G., López-Barajas MP, Mendoza L, Palma O, Velázquez O, Tapia R, Sepúlveda J. (2003). Encuesta Nacional de Salud 2000. Cuernavaca, Morelos, México.

Round 2
Reviewer 2 Report
Thank you for taking the time and effort to address my comments. This has improved the quality of the manuscript.
I noticed a few details that need correcting:-
-In text-citation styles are inconsistent "publication by Reinehr and Roth [4]" versus "Catillo-López et al. assessed"
Line 58- I think should be 'linked to' rather 'link to'
Line 63- 'a nation-wide research of a decreasing'- please remove the 'a'
Lines 72-73- 'The number of households 72 was allocated proportionally per urban and rural area' please replace was with were
Line 95- A full-stop is missing
Author Response
Comments and Suggestions for Authors
Thank you for taking the time and effort to address my comments. This has improved the quality of the manuscript.
I noticed a few details that need correcting:-
-In text-citation styles are inconsistent "publication by Reinehr and Roth [4]" versus "Catillo-López et al. assessed"
Corrected (line 43)
Line 58- I think should be 'linked to' rather 'link to'
Corrected (line 58)
Line 63- 'a nation-wide research of a decreasing'- please remove the 'a'
Removed (line 63)
Lines 72-73- 'The number of households 72 was allocated proportionally per urban and rural area' please replace was with were
We refer to the “number of households” as a whole and, therefore, as a single unit, we consider the use of a singular verb.
Line 95- A full-stop is missing
Full-stop has been added (line 95)